# Rare Case of Squamous Cell Carcinoma Arising from an Intraosseous Epidermal Cyst: A Diagnostic Challenge

**DOI:** 10.3390/diagnostics15243173

**Published:** 2025-12-12

**Authors:** Jiro Ichikawa, Kojiro Onohara, Tomohiro Inoue, Masanori Wako, Tetsuhiro Hagino, Kouhei Mitsui, Tomonori Kawasaki, Hirotaka Haro

**Affiliations:** 1Department of Orthopaedic Surgery, Interdisciplinary Graduate School of Medicine, University of Yamanashi, Chuo 409-3898, Yamanashi, Japan; wako@yamanashi.ac.jp (M.W.); tetsuhiro.hagino@gmail.com (T.H.); km.tykk@icloud.com (K.M.); haro@yamanashi.ac.jp (H.H.); 2Department of Diagnostic Radiology, Interdisciplinary Graduate School of Medicine, University of Yamanashi, Chuo 409-3898, Yamanashi, Japan; konohara@yamanashi.ac.jp; 3Department of Human Pathology, Interdisciplinary Graduate School of Medicine, University of Yamanashi, Chuo 409-3898, Yamanashi, Japan; tomohiroi@yamanashi.ac.jp; 4Department of Pathology, Saitama Medical University International Medical Center, Hidaka 350-1298, Saitama, Japan; tomo.kawasaki.14@gmail.com

**Keywords:** epidermal cyst, squamous cell carcinoma, pathology, differential diagnosis, magnetic resonance imaging

## Abstract

We report a rare case of squamous cell carcinoma (SCC) arising from an intraosseous epidermal cyst (EC) in the distal phalanx of the left thumb. A 76-year-old male presented with progressive thumb pain experienced over the previous six months. Radiography revealed a radiolucent lesion without marginal sclerosis, and magnetic resonance imaging showed peripheral contrast enhancement with no solid components. Surgery revealed a bone-originating mass without adhesion to the surrounding skin or nail bed, which histopathological findings determined contained both cystic epithelium with laminated keratin and invasive keratinizing tumor cells, confirming SCC arising from an intraosseous EC. No primary lesion or lymph node enlargement was identified by postoperative computed tomography. Although wide resection and chemotherapy were proposed, the patient declined further intervention beyond the curettage performed during surgery, opting for close observation only. No recurrence or metastasis has been observed in the five years since the surgery. Intraosseous ECs are extremely rare, with malignant transformation even more uncommon. Accurate diagnosis requires histopathological confirmation, as imaging alone is insufficient. This case highlights the importance of considering intraosseous EC in the differential diagnosis of bone lesions and underscores the need for further case accumulation to clarify optimal management strategies.

**Figure 1 diagnostics-15-03173-f001:**
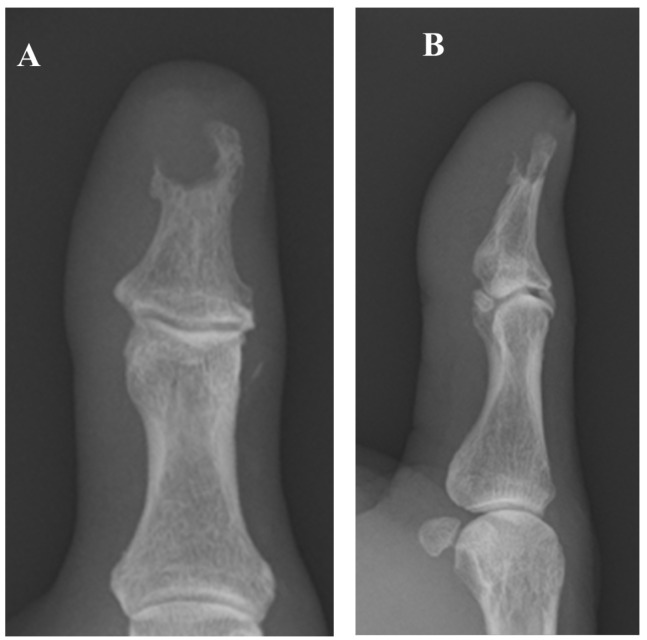
The patient is a 76-year-old man who began experiencing pain in his left thumb approximately six months prior to his initial visit to our hospital. Worsening pain over the past month had prompted him to consult a local physician. Radiography revealed a bone tumor, leading to his subsequent referral to our facility. At his initial visit, the patient presented with spontaneous pain and tenderness in the left thumb but exhibited no sensory deficit, circulatory disturbance, or restricted range of motion. The medical history of the patient included oral treatment for diabetes, which was initiated 25 years previously, and coronary stent placement following myocardial infarction 20 years previously; there was no history of trauma to the left thumb. Laboratory tests revealed mild renal impairment (estimated glomerular filtration rate, 42 mL/min/1.73 m^2^ [normal range, ≥60 mL/min/1.73 m^2^]), anemia (hemoglobin, 12.8 g/dL [normal range, 13.7–16.8 g/dL]), and a slightly elevated glycated hemoglobin level (6.1% [normal range, 4.9–6.0%]). The white blood cell count and C-reactive protein level were within normal limits, with no signs of inflammation. (**A**,**B**) Plain radiographs showed a radiolucent lesion in the distal phalanx without marginal sclerosis; no periosteal reaction or internal calcification was observed.

**Figure 2 diagnostics-15-03173-f002:**
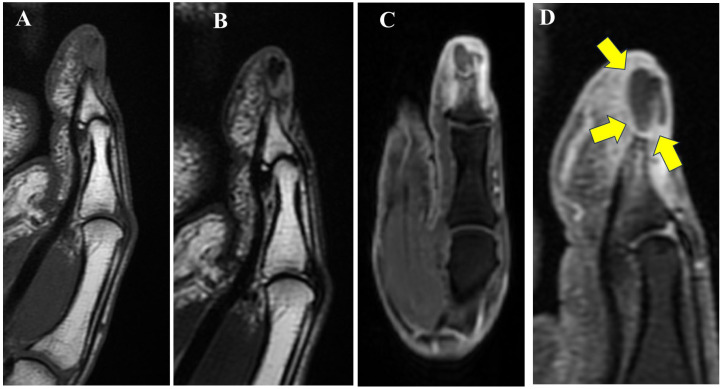
Magnetic resonance imaging (MRI) revealed that the lesion demonstrated (**A**) low signal intensity on T1-weighted imaging (T1WI) compared to normal bone marrow, (**B**) faint low signal on T2-weighted imaging (T2WI), and (**C**) heterogeneous high signal intensity on fat-suppressed T2WI. (**D**) Contrast enhancement was observed only at the periphery of the mass, as indicated by the yellow arrows. Additionally, areas of low signal on T1WI and high signal on fat-suppressed T2WI with contrast enhancement were observed in the surrounding bone marrow and subcutaneous tissue. Based on these imaging findings, the differential diagnoses included intraosseous epidermal cyst (EC), giant cell reparative granuloma, and metastatic bone tumor. An incisional biopsy was therefore performed to establish a definitive diagnosis.

**Figure 3 diagnostics-15-03173-f003:**
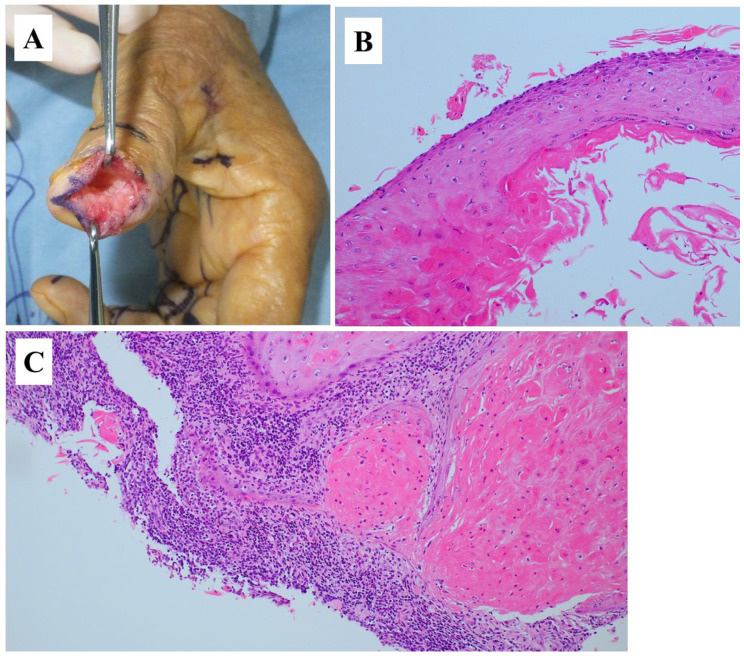
(**A**) Subcutaneous dissection revealed a mass protruding from within the bone, without adhesion to the surrounding skin or nail bed. Upon incising the capsule, the internal contents were identified and submitted for intraoperative rapid pathological examination. Although a definitive diagnosis of EC could not be made at that time, the lesion was excised with care to ensure the complete removal of the capsule. Histopathological examination of hematoxylin and eosin-stained sections revealed two distinct components: (**B**) a cystic lesion lined by stratified squamous epithelium with laminated keratin (magnification, 100×), and (**C**) an area showing invasive proliferation of keratinizing tumor cells (magnification, 100×). Based on these findings, a diagnosis of squamous cell carcinoma (SCC) arising from an intraosseous EC was made. Postoperative chest–abdomen–pelvis computed tomography (CT) did not identify a primary lesion or lymph node enlargement. ECs are subcutaneous cystic lesions commonly encountered in routine clinical practice; however, intraosseous occurrence is extremely rare [1,2]. SCC arising from a subcutaneous EC is also uncommon, with reported incidences ranging from 0.011% to 2.2% [3,4]. To the best of our knowledge, this is the first reported case of SCC arising from an intraosseous EC. While subcutaneous ECs may occasionally invade adjacent bone tissue [1], in the present case the skin at the incision site was not directly continuous with the mass, supporting the diagnosis of a primary intraosseous lesion. Intraosseous ECs are reported to occur more frequently in males, with common sites including the distal phalanx and skull [2]. Although the pathogenesis remains unclear, the involvement of congenital factors, trauma, and iatrogenic factors has been proposed. Momeni et al. reviewed seven cases of ECs arising in the distal phalanx and found a history of trauma in all of them [5]. Adhesion between the periosteum of the distal phalanx and subungual tissues has also been proposed as a contributing factor [2]. Previous studies have demonstrated that patients with SCC arising from subcutaneous ECs have a mean age of 62.8 (range, 28–96) years; 69% are male. Reported tumor sizes and durations from initial appearance to diagnosis range from 0.7 to 20 (mean, 5) cm and 0.5 to 480 (median, 92.6) months, respectively. Common symptoms include rapid growth (48.6%), pain (24.2%), and cutaneous changes such as discharge, ulceration, or skin breakdown (38.2%) [3]. The mechanism underlying the malignant transformation of an EC to SCC remains unclear, although chronic inflammation and infection have been proposed as potential triggers [3]. However, these factors were not observed in the present case, in which the absence of trauma history, the relatively short period from symptom onset to diagnosis, and minimal cutaneous changes were considered more consistent with an intraosseous EC. Radiography of intraosseous ECs typically reveal radiolucent lesions, cortical defects, or non-thinned cortices [2], consistent with the findings of the present case and also observed in benign bone tumors. Previous studies have reported that subcutaneous ECs are associated with complex MRI findings, with mixed low and high signal intensities in 50% of cases on T1WI and in 58% of cases on T2WI [6]. Contrast-enhanced MRI typically shows enhancement along the peripheral rim but not internally [6]. Key imaging features that help differentiate SCC from EC include cyst wall thickness and the presence of solid components [4]; however, these features were not prominent in the present case and SCC was therefore not initially suspected. A previous review of 37 bone tumors arising in the distal phalanx reported that the most common benign tumors were enchondromas and ECs; of the three malignant tumors, two were metastatic and one was SCC with direct invasion [7]. Given this background, imaging findings lead to a broad differential diagnosis, and histopathological examination is essential for accurate diagnosis. Additionally, metastatic bone tumors should be excluded based on imaging results and clinical context, with CT or positron emission tomography-CT considered. Currently, the optimal treatment strategies for SCC arising from intraosseous ECs remain unclear. Wide excision is the standard treatment for SCC arising from subcutaneous ECs [3], and a similar approach may therefore be appropriate for intraosseous SCC. In the present case, curettage was performed during the initial surgery. Although additional wide resection and chemotherapy were proposed following diagnosis, the patient declined further intervention and was instead placed under close follow-up. There has been no evidence of recurrence or metastasis in the five years since the surgery. However, further accumulation of cases is needed to establish effective treatment strategies.

## Data Availability

The data presented in this study are available from the corresponding author upon reasonable request.

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
