# Peer review of "Rare Case of Squamous Cell Carcinoma Arising from an Intraosseous Epidermal Cyst: A Diagnostic Challenge"

_diagnostics, 2025, doi:10.3390/diagnostics15243173_

Round 1

Reviewer 1 Report (Previous Reviewer 2)

Comments and Suggestions for Authors

Thank you for providing the revised manuscript under discussion.

The authors have exchanged Figure 3b. The new figure depicts a squamous cell formation that can be interpreted as highly differentiated squamous cell carcinoma, although it deviates from typical characteristics.

In this situation, I believe that the editor-in-chief should make the final decision regarding the publication of the manuscript.

Thank you for sharing this intriguing case with me.

On behalf of Dr. Albert Roessner. 

Reviewer 2 Report (Previous Reviewer 3)

Comments and Suggestions for Authors

My comments has been introduced. I recommed to publish the manuscript in the present form.

This manuscript is a resubmission of an earlier submission. The following is a list of the peer review reports and author responses from that submission.

Round 1

Reviewer 1 Report

Comments and Suggestions for Authors

"Extremely Rare" This phrase does not sound scientific. Please revise. 

Please refrain from AI-generated phrases. If any LLM has been used for editing etc., please disclose. 

Abstract needs one clear aim sentence. 

Please recheck keywords in MeSH. 

Does each author meet first two criteria of ICMJE?

Author Response

Reviewer 1

1) "Extremely Rare" This phrase does not sound scientific. Please revise.

Response: Thank you for this comment. To address this point, we have changed “Extremely Rare” to “Rare Case” in the title (Line 2).

Please refrain from AI-generated phrases. If any LLM has been used for editing etc., please disclose.

Response: Thank you for your careful review of the manuscript. We can confirm that AI was not used in the preparation or editing of the manuscript. Instead, professional English editing was performed by Editage.

Please recheck keywords in MeSH.

Response: Thank you for this helpful suggestion. We have revised the keywords to include only MeSH terms (Lines 34-35).

Does each author meet first two criteria of ICMJE?

Response: Thank you for this comment. We can confirm that all authors meet the first two criteria of ICMJE and we have revised the Author Contributions section to clarify this point (Lines 115-117).

Reviewer 2

This is a case report on squamous cell carcinoma arising from an intraosseous epidermal cyst. Several issues are identified with the manuscript. Firstly, the histologic figures do not convincingly demonstrate malignancy. Figure 3 C does not reveal an invasive pattern. Secondly, the reviewer considers this simple case report with obvious flaws to not meet the scientific standards of the journal Diagnostics.

Therefore, rejection is recommended.

Response: Thank you for your thorough review of the manuscript. We agree that Figure 3C did not illustrate invasion as well as it might, and to address this issue we have changed the image used in Figure 3C.

Reviewer 3

The request for review of the manuscript concerns a text on an extremely rare pathology. Intraosseous carcinomas are very rare. Osteosarcoma is more common, although this connective tissue pathology is not common in orthopedics. Here, the authors show the development of squamous cell cancinoma inside the last phalanx of the hand, which arose in the bone cyst wall. The study is clearly written, very well documented, and testifies to the scientific and clinical experience of the authors.

I would only have one request for the authors. Maybe they could enlarge the Fig 2D images and use arrows to indicate the areas of peripheral contrast enhancement in T2-weight imaging. This is very interesting, diagnostically important, and important for treatment planning.I have only one request for the authors. Perhaps they could enlarge the Fig. 2D images and use arrows or outlines the locations of peripheral contrast enhancement. This is very interesting, diagnostically important, and crucial for treatment planning.

Response: Thank you for your comments and for the helpful suggestion. We agree that the areas of peripheral contrast enhancement in T2-weighted images could be better highlighted and we have therefore added arrows to the image as you recommended (Figure 2D and Lines 56-57).

Reviewer 2 Report

Comments and Suggestions for Authors

This is a case report on squamous cell carcinoma arising from an intraosseous epidermal cyst. Several issues are identified with the manuscript. Firstly, the histologic figures do not convincingly demonstrate malignancy. Figure 3 C does not reveal an invasive pattern. Secondly, the reviewer considers this simple case report with obvious flaws to not meet the scientific standards of the journal Diagnostics.

Therefore, rejection is recommended.

Author Response

(The authors gave the same response as above.)

Reviewer 3 Report

Comments and Suggestions for Authors

The request for review of the manuscript concerns a text on an extremely rare pathology. Intraosseous carcinomas are very rare. Osteosarcoma is more common, although this connective tissue pathology is not common in orthopedics. Here, the authors show the development of squamous cell cancinoma inside the last phalanx of the hand, which arose in the bone cyst wall. The study is clearly written, very well documented, and testifies to the scientific and clinical experience of the authors. 

I would only have one request for the authors. Maybe they could enlarge the Fig 2D images and use arrows to indicate the areas of peripheral contrast enhancement in T2-weight imaging. This is very interesting, diagnostically important, and important for treatment planning.I have only one request for the authors. Perhaps they could enlarge the Fig. 2D images and use arrows or outlines the locations of peripheral contrast enhancement. This is very interesting, diagnostically important, and crucial for treatment planning.

Author Response

(The authors gave the same response as above.)
